# 4-Oxatricyclo[5.2.1.0^2,6^]dec-8-ene-3,5-dione Derivatives as NMDA Receptor- and VGCC Blockers with Neuroprotective Potential

**DOI:** 10.3390/molecules25194552

**Published:** 2020-10-05

**Authors:** Ayodeji O. Egunlusi, Sarel F. Malan, Sylvester I. Omoruyi, Okobi E. Ekpo, Jacques Joubert

**Affiliations:** 1Pharmaceutical Chemistry, School of Pharmacy, University of the Western Cape, Private Bag X17, Bellville 7535, South Africa; ayodeji86@gmail.com (A.O.E.); sfmalan@uwc.ac.za (S.F.M.); 2Department of Medical Biosciences, University of the Western Cape, Private Bag X17, Bellville 7535, South Africa; 3405455@myuwc.ac.za (S.I.O.); oekpo@uwc.ac.za (O.E.E.)

**Keywords:** neurodegenerative disorders, 4-oxatricyclo[5.2.1.0^2,6^]dec-8-ene-3,5-dione, cytotoxicity, neuroprotection, NMDA receptor, voltage gated calcium channels, polycyclic cages

## Abstract

The impact of excitotoxicity mediated by *N*-methyl-D-aspartate (NMDA) receptor overactivation and voltage gated calcium channel (VGCC) depolarization is prominent among the postulated processes involved in the development of neurodegenerative disorders. NGP1-01, a polycyclic amine, has been shown to be neuroprotective through modulation of the NMDA receptor and VGCC, and attenuation of MPP^+^-induced neurotoxicity. Recently, we reported on the calcium modulating effects of tricycloundecene derivatives, structurally similar to NGP1-01, on the NMDA receptor and VGCC of synaptoneurosomes. In the present study, we investigated novel 4-oxatricyclo[5.2.1.0^2,6^]dec-8-ene-3,5-dione derivatives for their cytotoxicity, neuroprotective effects via attenuation of MPP^+^-induced neurotoxicity and calcium influx inhibition abilities through the NMDA receptor and VGCC using neuroblastoma SH-SY5Y cells. All compounds, in general, showed low or no toxicity against neuroblastoma cells at 10–50 µM concentrations. At 10 µM, all compounds significantly attenuated MPP^+^-induced neurotoxicity as evident by the enhancement in cell viability between 23.05 ± 3.45% to 53.56 ± 9.29%. In comparison to known active compounds, the derivatives demonstrated mono or dual calcium modulating effect on the NMDA receptor and/or VGCC. Molecular docking studies using the NMDA receptor protein structure indicated that the compounds are able to bind in a comparable manner to the crystallographic pose of MK-801 inside the NMDA ion channel. The biological characteristics, together with results from in silico studies, suggest that these compounds could act as neuroprotective agents for the purpose of halting or slowing down the degenerative processes in neuronal cells.

## 1. Introduction

Glutamate is a primary excitatory neurotransmitter that plays a prominent role in various physiological functions such as synaptic plasticity, learning, memory and other cognitive processes. In order to fulfil these functions, there is a need to constantly maintain low concentrations of extracellular glutamate in both neuronal and astroglial cells [1,2]. Under certain pathological conditions such as Alzheimer’s disease (AD), Parkinson’s disease (PD), Amyotrophic lateral sclerosis (ALS) and Huntington’s disease (HD), there is an increase in the level of extracellular glutamate, which becomes toxic to the neuronal cells leading to excitotoxicity [2,3].

In excitotoxicity, excessive extracellular glutamate and membrane depolarisation over-activate NMDA receptor and voltage-gated calcium channels to increase intracellular calcium ions, leading to the disruption in calcium homeostasis [4,5]. Under normal physiological conditions, intracellular organelles such as the mitochondria and the endoplasmic reticulum are able to sequestrate Ca^2+^ and maintain homeostasis. However, these recovery processes are hindered due to prolonged glutamate activity leading to calcium overload [2,6,7]. Due to the sensitivity of neurons towards intracellular calcium ion concentration, consistent elevation of intracellular calcium ion levels induces a cascade of events via a number of enzymatic pathways [8], leading to mitochondrial dysfunction, calpain activation, lipid peroxidation, oxidative stress and protein aggregation [9]. These processes consequently damage cellular components and organelles, ultimately leading to neuronal cell death. Therefore, molecules capable of modulating calcium influx via the NMDA receptor and/or the VGCC could provide protection against calcium overload mediated by prolonged glutamate activity.

Over the past few decades, uncompetitive NMDA receptor channel blockers such as dizolcipine (MK-801) and phencyclidine (PCP) (Figure 1) have been extensively studied. However, they are marked by undesirable psychotomimetic side effects such as hallucination, psychosis, dysphoria and amnesia and are thus unsuitable for clinical purposes. On the other hand, adamantanes, such as amantadine and memantine (Figure 1), are clinically well tolerated for the treatment of Parkinson’s disease and Alzheimer’s disease, respectively, due to their atypical mechanism of action on the NMDA receptors [10,11,12,13]. In contrast to MK-801, they exhibit moderate affinity towards the phencyclidine binding site of the NMDA receptor channels to effectively block the NMDA receptor while demonstrating minimal adverse effects [13]. Like memantine, the structurally related compound, NGP1-01, has been shown to uncompetitively block the NMDA receptor. Additionally, NGP1-01 has also been shown to inhibit *L*-type VGCC and is effective against 1-methyl-4-phenylpyridinium (MPP^+^)-induced neurotoxicity. NGP1-01 is thus a promising multifunctional neuroprotective agent as demonstrated by several studies [14,15,16,17,18,19,20,21,22,23].

In a previous study, the calcium modulating effects of a small series of tricycloundecene derivatives via the NMDA receptor and VGCC were reported (Figure 2). The significant activity observed for tricyclo[6.2.1.0^2,7^]undec-9-ene-3,6-dione (**1**) was attributed to its structural similarities to MK-801 and the adamantanes [24]. In the present study, a series of novel 4-oxatricyclo[5.2.1.0^2,6^]dec-8-ene-3,5-dione derivatives (**2**–**13**, Figure 3), structurally related to **1**, were investigated for their cytotoxic profile, ability to block calcium influx via the NMDA receptor and VGCC, and neuroprotective activities through MPP^+^-induced neurotoxicity using SH-SY5Y neuroblastoma cells. The synthesis and characterization of these compounds were recently described by our group [25]. These derivatives contain a number of aliphatic, heterocyclic or aromatic functionalities that are expected to provide valuable information on the structure activity relationships of this group of compounds. Finally, these derivatives were also investigated for their physicochemical and pharmacokinetic properties using in silico models. Due to their structural similarities with NGP1-01, MK-801, the adamantanes and **1**, we postulate that these derivatives would exhibit similar or improved multifunctional neuroprotective abilities.

## 2. Results and Discussion

### 2.1. Biological Studies

#### 2.1.1. Cytotoxicity Studies

The cytotoxicity profiles of compounds **2**–**13** were assessed by screening the test compounds at 10 μM, 50 μM, and 100 μM against the human neuroblastoma SH-SY5Y cell line. This was determined by exposing treated cells to each test compound at the specified concentrations for 24 h and subsequently measuring the percentage cell viability using the MTT assay [26]. Compared to the untreated controls, all compounds, except compounds **2**, **3**, **9** and **10**, showed insignificant toxicity towards the cells at 10 µM (Figure 4).

In general, all compounds except **3**, **6** and **10**, showed a decrease in cell viability as the concentrations increased. The increase in cell viability with an increase in concentration, as observed for **3**, **6** and **10**, may be due to these molecules being able to reduce baseline apoptotic processes within the cells or show proliferative effect in the neuroblastoma cells, especially compound **6**. Additional studies would be necessary to better define the effects observed.

At 100 µM, virtually all compounds with the exception of **3**, **6** and **10** exhibit statistically significant toxicity. Compound **2** was the most cytotoxic towards neuroblastoma cell as indicated by the percentage cell viability of 51.02 ± 0.86%. However, the incorporation of various amine moieties (**3**–**13**) led to a significant (*p* < 0.05) increase in percentage cell viability compared to **2**. In comparison to compound **8** (69.77 ± 6.21%), the replacement of the phenylhydrazine moiety with a benzylamine moiety (**3)** resulted in a slight restoration of cells as showed by cell viability of 86.98 ± 6.33%. However, conversion of the benzylamine moiety to a propargylamine moiety significantly restored cell viability to 127.19 ± 2.94% (**6**). Likewise, the aza substitution of the primary amine in compound **9** (79.96 ± 1.06%) led to an increase in cell viability of 93.05 ± 7.25% (**10**). However, little or no change was observed with *N,N*-dimethylation as seen in compound **13** (76.03 ± 5.18%). Due to the minimal cytotoxicity observed at 10 µM, further neuroprotection and calcium influx assays were conducted at this concentration as it should not significantly affect the viability of the neuroblastoma cells.

#### 2.1.2. Neuroprotection Studies

SH-SY5Y neuroblastoma cells are known to exhibit physiological and pathological similarities with primary neurons, thus it usage in screening known and unknown calcium blockers. Additionally, NMDA induced SH-SY5Y cell injury model is believed to mimic glutamate-induced toxicity in human neurons and is generally accepted as activity screening model for the neuroprotective effect of NMDA receptor inhibitors [27,28]. In this study, the ability of compounds **1**–**13** to protect neuroblastoma cells against MPP^+^-induced toxicity was assessed using the MTT assay as reported in the literature [26]. 1-Methyl-4-phenylpyridinium (MPP^+^), a neurotoxin, is an active metabolite of 1-methyl-4-phenyl-1,2,3,6-tetrahydropyridine (MPTP) derived from enzymatic action of monoamine oxidase B (MAO-B) in the inner mitochondrial membrane. MPP^+^, as opposed to MPTP, is an ideal toxin in a neurotoxicity study where neuroblastoma SH-SY5Y cells are ultilized, owing to the inability of SH-SY5Y cells to metabolize MPTP. It has been shown to evoke 50% neuronal cell death by inhibiting respiratory complex 1 of the mitochondrial electron transport chain, thus justifying its usage in this study [29,30,31]. Compared to the control (DMSO only), exposure to 2 mM of MPP^+^, over a period of 24 h, resulted in a significant loss of cells as demonstrated by 49.89 ± 2.11% cell viability [30,31]. However, pre-treatment of cells with 10 µM of compounds **1**–**13** for 2 h and subsequent treatment with MPP^+^ inhibited neuronal cell damage or loss as demonstrated by a significant enhancement in cell viability between 23.05 ± 3.45% to 53.56 ± 9.29% (Table 1) when compared to the MPP^+^ only treated cells.

The conversion of the dione ring from a 6-membered ring (**1**) to a 5-membered ring (**2**) showed no significant change in neuroprotective activity. However, the incorporation of a benzylamine conjugate to compound **2** (30.87 ± 3.1) significantly improved the neuroprotective activity of **3** (53.56 ± 9.3). This could be attributed to the increased lipophilic nature or inclusion of the nitrogen, which may have allowed better interaction with protein targets involved in the development of neurodegeneration. The replacement of the benzylamine moiety (**3**) with a phenylhydrazine moiety (**8**) and the opening of the dione ring in **3** to form the diamide derivative (**4**) led to reduction in cell viability of 35.32 ± 5.65% and 29.06 ± 3.02%, respectively. In comparison to compound **2**, there was little or no change in the neuroprotective activities of 1-aminopiperidine (**5**; 32.08 ± 8.19%), propargylamine (**6**; 27.53 ± 6.04%) and propylamine (**7**; 30.46 ± 0.77%) moieties, thus suggesting the tolerability of these conjugates. The neuroprotective activity of the diamine conjugate (**9**; 29.92 ± 6.77%) was similar to compound **2**. The addition of a CH_2_ group, as observed in **11**, resulted in a slight improvement of 35.11 ± 1.12%. A further increase in chain length led to a slight decrease in activity (**12**; 28.75 ± 10.44%). In comparison to **9**, dimethyl- (**13**) and aza- (**10**) substitutions at the primary nitrogen group led to a slight reduction in cell viability of 23.05 ± 3.45% and 27.18 ± 8.56%, respectively. Despite the slight reduction in activity, the enhancement in cell viability of compounds **10** and **13**, when compared to MPP^+^ only treatment cells, suggest lipophilic interactions rather than nitrogen-protein interactions.

#### 2.1.3. NMDA Receptor

##### NMDA Receptor Mediated Ca^2+^ Studies

The calcium blocking effects of compounds **1**–**13** were assessed by measuring the increase in intracellular calcium ions through the NMDA receptor on SH-SY5Y neuroblastomas cells in a Fura-2 based assay. Figure 5 illustrates the fluorescence profiles or behaviors of the cells before and after injection of the stimulating buffer (NMDA/glycine) in the presence and absence of the positive control (MK-801). When compared to the control (DMSO), a change in fluorescence intensities were observed after NMDA receptors activation, at 11.1 s, for MK-801, which corresponds to its calcium influx profile. The calcium modulating effects of the test compounds were determined by monitoring the change in fluorescence intensities relative to the control (DMSO). At 10 µM, MK-801 and NGP1-01 significantly reduced calcium influx via the NMDA receptor by 47.51 ± 2.34% and 38.15 ± 3.69%, respectively, when compared to the control (DMSO only). In comparison, compounds **1**–**13** displayed calcium influx reduction of 23.47 ± 3.47% to 64.27 ± 1.24% with compounds **1**, **7**, and **13** showing activities exceeding that of NGP1-01 and MK-801 (Table 1).

The conversion of the 6-membered dione ring (**1**) to a 5-membered ring (**2**) significantly diminished the calcium blocking activity from 51.00 ± 3.22% to 23.47 ± 3.47%. This decrease in activity could be attributed to the less lipophilic- and/or hydrogen bonding character of compound **2**, which could have minimized its interaction with the NMDA receptor. However, the introduction of a benzylamine moiety to **2** significantly improved the calcium blocking ability to 43.30 ± 1.68% (**3**). The replacement of the benzylamine moiety with 1-aminopiperidine (**5**; 43.68 ± 0.46), propargylamine (**6**; 43.75 ± 2.19%) and phenylhydrazine (**8**; 45.11 ± 4.30%) moieties showed no significant change in activities when compared to **3**. Compared to the propargylamine conjugate (**6**), compound **7** (propylamine conjugate) displayed a slight improvement in calcium influx reduction (53.90 ± 2.09%), which may be linked its decreased conformational rigidity leading to enhanced interactions with the NMDA receptor.

The introduction of a diamine conjugate (**9**) led to a slight improvement in calcium inhibition of 36.82 ± 3.36%, when compared to compound **2**. The activity of **9** is attributed to the presence of the basic amino functionality which could have improved its interaction with the NMDA receptor binding site. Addition of a CH_2_ group to the diamine of **9** resulted in a significant increase in calcium influx blockage of 50.40 ± 2.28% (**11**) and a further increase in chain length displayed no significant change in calcium reduction as seen in compound **12**. The increase in the activities of **11** and **12** could be attributed to their ability to reach a hydrophobic pocket within the NMDA receptor binding site leading to better interactions. Interestingly, dimethyl- (**13**) and aza- (**10**) substitution of the primary amine of **9** led to a significant increase in calcium influx blockage of 64.27 ± 1.24% and 50.40 ± 2.28%, respectively. The activities of **10** and **13** could be attributed to increased lipophilicity [32].

##### NMDA Receptor Docking Studies

In order to obtain structural insights regarding the binding interactions and orientation of compounds **2**–**13** as NMDA receptor antagonists, we performed molecular docking studies using the X-ray crystal structure of the NMDAR co-crystallized with MK-801 (PDB code: 5UN1, www.rcsb.org). Molecular Operating Environment (MOE) software [33] was used to conduct the docking experiment and score the molecules according to their binding affinity. As compounds **2**–**13** have structural similarities to MK-801, they are expected to bind and interact with the NMDA receptor in a similar manner [34]. To confirm if the docking parameters and computational procedure could reproduce experimental results, the co-crystallized ligand, MK-801, was redocked. The docking pose obtained was similar to the co-crystallized one, with a RMSD of 0.63 Å and a binding affinity of −8.87 kca·mol^−1^. To further validate the docking protocol, four known NMDA receptor antagonists binding at the same site as MK-801, were also docked. The molecules included phencyclidine [35], R-ketamine [35], memantine [36] and amantadine [37]. The results are summarized in Table 2 and illustrates a clear correlation between binding affinity and experimental Ki value. 

The results from the docking experiment revealed that all the molecules were able to bind in a comparable position and manner to the crystallographic pose of MK-801 inside the NMDAR ion channel. Figure 6A shows the binding of representative molecules (**3**, **8** and **13**) within the MK-801 binding site of the NMDA receptor. The molecules were shown to form contacts with nearby “greasy” hydrophobic residues (e.g., Ala, Val, Met, Leu) similarly to MK-801 (Figure 7). In addition, **2**–**13** presented with binding affinities (−5.89 to −8.31 kca·mol^−1^) that were within the experimental binding affinity range of the known NMDA receptor antagonists (Table 2), suggesting comparable antagonistic activities. To further confirm the similar binding of the compounds compared to MK-801, a flexible alignment experiment was executed in MOE (Figure 6B). The results indicate that, in general, the compounds are able to assume conformations that closely follow the V-shaped conformation of the co-crystallized MK-801. These docking studies therefore clearly reveals that the significant NMDA receptor channel blocking abilities observed for **2**–**13**, could be attributed to their similar binding to and blocking of the NMDA receptor channel as MK-801.

#### 2.1.4. Voltage Gated Ca^2+^ Studies

The measurement of Ca^2+^ influx blockage by the test compounds, after VGCC depolarization (KCl, 770 mM), were conducted in the same manner as the NMDA receptor mediated Ca^2+^ studies. In the absence of a depolarizing buffer, the VGCC are closed and impermeable to calcium ions. However, injection of the buffer at 11.1 s changed the fluorescence profiles or behavior of the neuroblastoma cells. The fluorescence intensities obtained from the ratio (340/380) corresponds to the VGCC mediated calcium influx profile of the control or test sample (nimodipine). In comparison to the control (DMSO), nimodipine displayed a cell fluorescence profile that suggest a decreased fluorescence intensity in the presence of calcium (Figure 8). The calcium modulating effects of test compounds (NGP1-01, **2**–**13**) were determined by assessing their fluorescence intensities relative to the control. At 10 µM, nimodipine (known calcium channel blocker) and NGP1-01 showed VGCC inhibition of 34.07 ± 4.78% and 34.20 ± 3.95%, respectively. In comparison to the known blockers, all compounds (**1**–**13**) displayed moderate to good VGCC inhibitions of 7.46 ± 4.58% to 37.94 ± 9.27%, with compounds **3** and **11** exceeding the activities of nimodipine and NGP1-01. The tricyloundecene derivative **1** showed VGCC inhibition of 25.36 ± 2.76%, and conversion to a 5-membered dione ring (**2**) led to an insignificant change in calcium blockage of 23.14 ± 8.34%. However, the incorporation of a benzylamine moiety (**3**) resulted in a slight improvement in activity to 37.20 ± 10.02% VGCC. The increased activity of **3** is linked to its enhanced lipophilicity, which may have improved its interactions with the VGCC. The replacement of the benzylamine moiety with a more basic phenylhydrazine moiety (**8**) diminished its calcium influx inhibitory effect (26.80 ± 2.42%). In comparison to **8**, compound **5**–**7** and **9** displayed little or no change in VGCC blockage, thus suggesting the tolerability of 1-aminopiperidine, propargylamine, propylamine and ethylenediamine moieties as VGCC blockers. The addition of a CH_2_ group to the diamine of **9** resulted in an improvement in VGCC inhibition of **11** (37.94 ± 9.27%), but a further increase in chain length led to a significant reduction in activity as seen in compound **12** (15.36 ± 3.53%). Lastly, the dimethyl substitution of the primary amine in **9** resulted in diminished VGCC blocking ability of 7.49 ± 4.58 (**13**). However, replacing the dimethyl substituent with an aza-substituent, as seen in compound **10**, led to a significant increase in VGCC inhibition of 31.42 ± 5.21% (*p* < 0.05).

#### 2.1.5. In Silico Pharmacokinetic and Drug-Likeness Evaluation

As a rule, a good drug candidate must not only possess good biological activities, but also acceptable pharmacokinetic profiles. Using a series of web-based tools, it is possible to predict the pharmacokinetics, ADME (absorption, distribution, metabolism, excretion) and drug-likeness of new chemical entities [38,39,40,41]. Early assessment of these parameters minimize pharmacokinetic related failures during drug development, thus the need for their consideration. The blood brain barrier (BBB) is an integral part of the brain that protect it through a physical barrier (tight junctions in endothelial cells preventing paracellular penetration) and a biochemical barrier (enzymes and active efflux). Therefore, data on BBB permeation of potential neuroprotective agents are of utmost importance [42]. In this study, the SwissADME web tool [42] was used to predict the drug-likeness and pharmacokinetic properties of amantadine, memantine and **1**–**13**. The adamantanes were included as reference compounds, because they are approved drugs for the treatment of neurodegenerative disorders [10,11,12,13]. The data generated are presented in Figure 9 and Table 3.

The pink area of the bioavailability radar (Figure 9) depicts the optimum range for lipophilicity (LIPO), size, polarity (POLAR), solubility (INSOLU), flexibility (FLEX), and saturation (INSATU), while the red lines present the predicted physicochemical properties for each compound [39,40]. It is designed to allow quick appraisal of drug-likeness. Similar to the adamantanes, the six predicted physicochemical properties for compounds **1**–**13** were located within the pink area of the radar plot, thus suggesting high gastrointestinal absorption and good oral bioavailability. This was confirmed by the GIA (high) data and the high bioavailability scores (0.55) observed for all compounds as illustrated in Table 3. Additionally, the BBB dataset predicted that all compounds except **6**, **9** and **10** would easily penetrate the blood brain barrier, making them good candidates as potential neuroprotective agents. Lastly, all compounds were found to obey the drug-likeness rules set by pioneer pharmaceutical companies; Veber’s (GSK), Lipinski’s (Pfizer), Egan’s (Pharmacia), Ghose’s (Amgen) and Muegge’s (Bayer) rules [40,43]. However, compounds **1** and **2** were found to exhibit one violation towards the Muegge rule, while compound **9** showed one violation towards the Ghose rule. In general, the results predict that these compounds (**1**–**13**) exhibit good bioavailability, sufficient BBB permeability and acceptable predicted pharmacokinetic profiles. These compounds could therefore be considered as excellent lead compounds or drug candidates for the treatment of neurodegenerative disorders.

## 3. Conclusions

The 4-oxatricyclo[5.2.1.0^2,6^]dec-8-ene-3,5-dione derivatives (**2**–**13**) displayed minimal or no cytotoxic effects against neuroblastoma cells and attenuated MPP^+^-induced toxicity with activities similar to the tricycloundecene derivative (**1**). These derivatives (**2**–**13**), co-treated with MPP^+^, displayed significant reduction in neuronal cell loss when compared to MPP^+^ only treated cells. The highest protection against MPP^+^-induced damaged was observed for compound **3** with a neuroprotective activity significantly exceeding that of compound **1**. The neuroprotective activities of this group of compounds suggest the influence of the lipophilic character and the inclusion of a primary nitrogen group. Compared to known NMDA receptor and VGCC blockers, all compounds (**1**–**13**) displayed moderate to good blockage of calcium influx at the NMDA receptor and/or VGCC. Compounds **7**, **10**, **11**, **12** and **13** displayed calcium inhibitory effects exceeding that of MK-801 and NGP1-01 in the NMDA receptor mediated Ca^2+^ studies, while compound **3** and **11** demonstrated calcium influx inhibition that exceed that of nimodipine and NGP1-01 in the voltage gated Ca^2+^ studies. Similar to NGP1-01 and compound **1**, the majority of the 4-oxatricyclo[5.2.1.0^2,6^]dec-8-ene-3,5-dione derivatives displayed dual calcium inhibition at the NMDA receptor and VGCC. This synergistic calcium influx blockage is expected to halt or significantly diminish calcium overload during excitotoxicity in the neurodegenerative process, thus offering protection to neuronal cells. The calcium blocking effects, in general, were more pronounced at the NMDA receptor than the VGCC. The structure activity relationships (SARs) of these compounds as NMDA receptor blockers suggest the influence of less conformational rigidity, basic amino functionality and lipophilicity, while the SARs as VGCC blockers indicate the importance of enhanced lipophilicity. In silico studies using the SwissADME online tool predicated acceptable gastrointestinal absorption, brain permeability and drug-likeness for all the derivatives and they can therefore be considered excellent lead compounds for further development. Despite the limited number of compounds for SARs, the findings from the biological- and in silico studies demonstrated promising neuroprotective abilities and drug-likeness profiles of these derivatives. Additionally, these results provide valuable information on the potential multifunctional neuroprotective abilities of 4-oxatricyclo[5.2.1.0^2,6^]dec-8-ene-3,5-dione derivatives and could assist in the design and development of structurally related compounds as neuroprotective agents.

## 4. Material and Method

### 4.1. Biological Studies

#### 4.1.1. Cell Line and Culture Condition

The human neuroblastoma SH-SY5Y cells were generously donated by the Blackburn Laboratory, University of Cape Town. Cells were grown in Dulbecco’s Modified Eagle Medium (DMEM, Gibco, Life Technologies Corporation, Paisley, UK), supplemented with 10% fetal bovine serum (FBS, Gibco, Life Technologies Corporation, Paisley, UK), 100 U/mL penicillin and 100 μg/mL streptomycin (Lonza Group Ltd., Verviers, Belgium). Cultures were incubated at 37 °C in humidified air with 5% CO_2_ and a medium change every three days. Cells were sub-cultured when they attained 70 to 80 percent confluency using a solution of 0.25% trypsin EDTA (Lonza Group Ltd., Verviers, Belgium).

#### 4.1.2. Cytotoxicity Studies

Confluent SH-SY5Y cells were seeded in a 96-well microtiter plastic culture plates at a density of 1 × 10^4^ cells/100 µL/well and incubated for 24 h. Thereafter, the culture media were replaced with fresh DMEM (90 µL) containing increasing concentrations (10 µM, 50 µM and 100 µM) of each test compounds (**2**–**13**) for 24 h. The control group of cells contained DMEM and DMSO in concentration similar to the highest concentration of the treated group and were processed and incubated concurrently with the treated group of cells. After treatment of cells, 10 µL of MTT (5 mg/mL) solution was added to obtain a final concentration of 0.5 mg/mL in each well and further incubated for 4 h at 37 °C. This was followed by addition of 100 µL of DMSO to ensure solubility of the formazan, and absorbance of the sample in each well was measured at 570 nm on a BMG Labtech Omega^®^ POLARStar multimodal plate reader. The absorbance data obtained were expressed as percentage of control, and all experiments were performed in triplicates. 

#### 4.1.3. Neuroprotection Studies

The SH-SY5Y cells were seeded in a 96 well plate at density of 1.0 × 10^4^ cells/100 µL/well. Plated cells were allowed to adhere to the bottom of the plastic plate. Thereafter cells were pre-treated with 10 µM of compounds (**1–13**) for 2 h before the addition of 2 mM of MPP^+^ and incubated for a period of 24 h. The 10 µM concentration was deemed appropriated as it showed minimal or no cytotoxic effect on cells. The untreated control cells were treated with DMEM which contained DMSO and were incubated in the same manner as the treated cells. After incubation for the set period of time, both treated and untreated cells were subjected to the MTT assay as described under Section 4.1.2.

#### 4.1.4. NMDA Receptor Mediated and VGCC Ca^2+^ Influx Studies

##### Cell Treatment

Confluent neuroblastoma SH-SY5Y cells were seeded in 100 μL of DMEM at a density of 1 × 10^5^ cells per well in a 96-well black plate. The seeded plate was incubated at 37 °C for 24 h so as to allow cell adherence to the plate bottom. After 24 h, 10 μL of Fura-2/AM (5 mM) was suspended in 9990 μL of fresh DMEM to obtain 5 μM final concentrations. The DMEM in each of the seeded wells was replaced with 100 μL of Fura-2/AM in DMEM suspension and was incubated for 1 h at 37 °C to allow diffusion of Fura-2/AM into the neuroblastoma cells. After 1 h, the Fura-2/AM in DMEM suspension was removed from each well, and the wells were washed with Krebs-HEPES solution (118 mM NaCl, 4.7 mM KCl, 20 mM HEPES, 30.9 mM glucose monohydrate) to remove any extracellular Fura-2/AM. Thereafter, 49 μL of calcium containing buffer (2 mM CaCl_2_.2H_2_O) was added to the wells and incubated at 37 °C for 30 min. The test compounds were added to the seeded wells for fluorescence analysis.

##### Measurement of Intracellular Calcium for NMDA-Mediated Studies

A stock solution of 0.5 mM of each test compound was prepared in an Eppendorf vial by dissolving an appropriate amount in 1 mL DMSO. One microliter of the test compounds was pipetted from the stock solutions and added to the treated cells in order to achieve final concentrations of 10 μM in each well. 1 μL of DMSO was used as negative control. Once sample preparation was completed, the 96-well plate was placed in a Synergy™ Mx monochromator-based fluorescent microplate reader (BioTek^®^, Germany) connected to Gen 5™ data analysis software. In the machine, the plate was further incubated at 37 °C for 30 min with intermittent shaking every minute before fluorescence reading. Calcium influx for each cell well was monitored over a period of 35 sec with 0.5 sec interval between wells. During fluorescence readings, 10 μL of stimulating buffer (4.7 mM KCl, 20 mM HEPES, 30.9 mM glucose monohydrate, 0.1 mM CaCl_2_.2H_2_O, 0.55 mM glycine, and 0.55 mM NMDA) was injected into each well after 10 sec to activate the NMDA receptors for purpose of calcium influx. The sample was exposed to dual wavelengths of 340 and 380 nm for the purpose of excitation, and fluorescence from emission at 510 nm was measured. The ratio of fluorescence (340/380) values after stimulation corresponds to the concentration of NMDA-mediated calcium influx within the 35 sec period. Each experiment was conducted in triplicate. The calcium influx of each test compound was expressed as percentage relative to the control (DMSO only) represented as 100% calcium influx. Percentage values were obtained from a fluorescence ratio (340/380 nm) versus time analysis using Microsoft Excel and graphs were generated in GraphPad Prism version 6.01 for windows.

##### Measurement of Intracellular Calcium for VGCC Studies

The same methods and calculations were used as the NMDA mediated Ca^2+^ influx studies, except that the NMDA receptor stimulating solution was replaced by a VGCC depolarizing buffer (5.4 mM NaCl, 770 mM KCl, 20 mM HEPES, 1.4 mM CaCl_2_.2H_2_O, 0.9 mM MgSO_4_, 5.5mM glucose monohydrate, 0.6 mM KH_2_PO_4_, 0.6 mM Na_2_HPO_4_, 10 mM NaHCO_3_).

##### Calculation of Percentage Calcium Inhibition in NMDA-Mediated and VGCC Studies

The fluorescence behavior of cells in the presence or absence of test samples was a reflection of its fluorescence intensities. These intensities, determined by the fluorescence ratio at 340 nm and 380 nm, were proportional to the intracellular calcium ion concentration upon depolarization of membrane or stimulation of NMDA receptors. The fluorescence ratios (340/380) versus time were plotted on a graph to derive a linear equation for the control (DMSO) or test compounds. The estimated calcium influx values were then extrapolated from the equation at 36.8 s. Relative to the control, the percentage calcium inhibition for all test compounds were calculated using the following formula:$$\%\;Inhibition\;=\;\frac{Control-Test}{Control}\times 100$$

#### 4.1.5. Statistical Analysis

Statistical analyses were performed on a GraphPadPrism software version 6.01 for windows (GraphPad software, LA Jolla, CA, USA), using the one-way analysis of variance (ANOVA) and *t*-test (unpaired). Statistically significant differences were set at *p* < 0.05.

### 4.2. Molecular Docking Studies

Molecular docking was carried out using the NMDA receptor crystal structure co-crystallized with MK-801 (PDB ID: 5UN1), which was recovered from the Brookhaven Protein Database (www.rcsb.org/pdb). Docking simulations were performed on the test compounds using Molecular Operating Environment (MOE) software [31] with the following protocol. (1) The NMDA receptor structure was checked for missing atoms, bonds and contacts. (2) Hydrogens and partial charges were added using the protonate 3D application in MOE. (3) The ligands were constructed using the builder module and were energy minimized using the MMFF94x force field. (4) Ligands were docked within the MK-801 binding site using the MOE Dock application, the poses were generated by the Triangle Matcher placement method. (5) The lowest energy ranked pose for each compound was retained and the interactions with binding pocket residues were analyzed. The flexible alignment module of MOE was used to study the conformational space of the molecules using a flexible description, and default parameters.

### 4.3. In Silico Pharmacokinetic and Drug-Likeness Evaluation

To analyze the drug-likeness and pharmacokinetic profiles of compounds **1**–**13**, the SwissADME online tool was utilized. All structures were drawn in ChemSketch 2.0 (2018) and saved as MDL molfiles. These molfiles were converted to SMILES (simplified molecular-input line-entry system format, which were ran on the web tool [39].

## Figures and Tables

**Figure 1 molecules-25-04552-f001:**
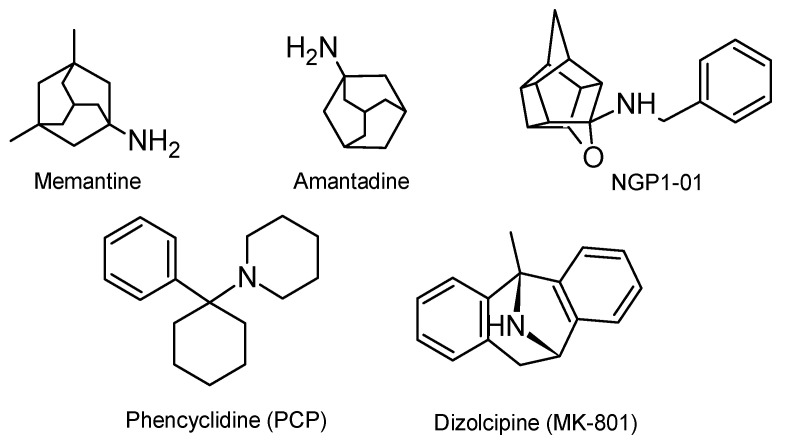
Uncompetitive NMDA receptor antagonists.

**Figure 2 molecules-25-04552-f002:**
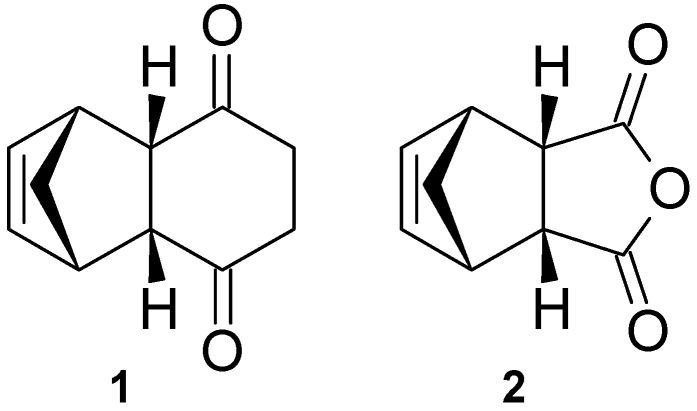
Tricyclo[6.2.1.0^2,7^]undec-9-ene-3,6-dione- (**1**) and 4-oxatricyclo[5.2.1.0^2,6^]dec-8-ene-3,5-dione (**2**) scaffolds.

**Figure 3 molecules-25-04552-f003:**
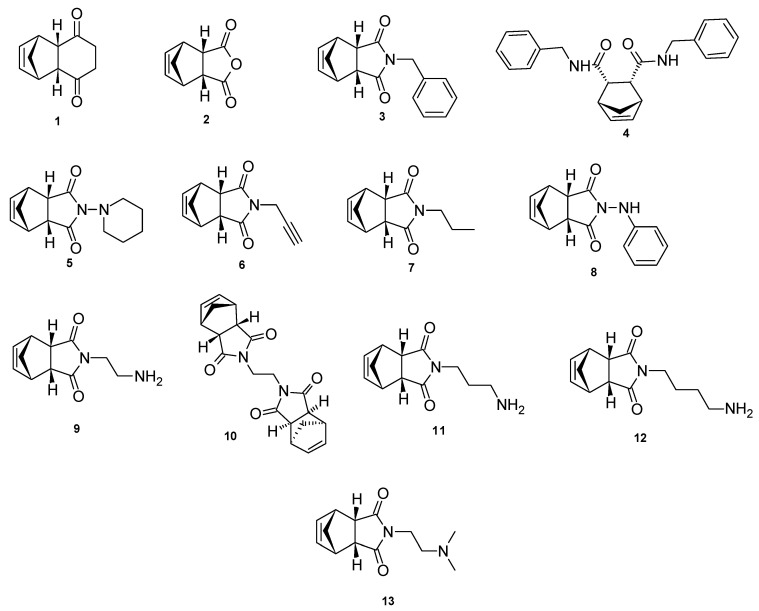
4-Oxatricyclo[5.2.1.0^2,6^]dec-8-ene-3,5-dione derivatives evaluated in this study.

**Figure 4 molecules-25-04552-f004:**
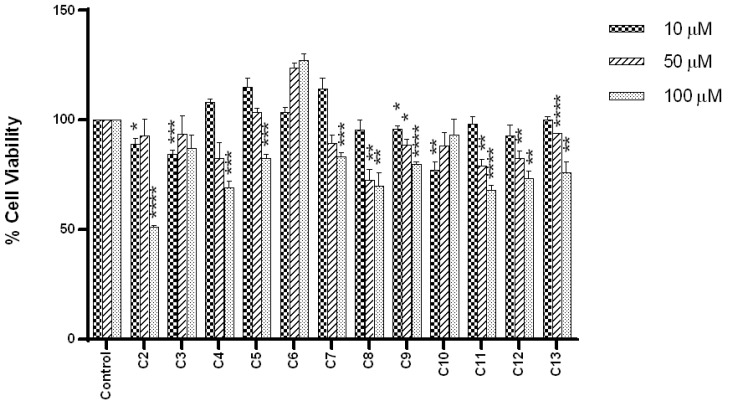
Cytotoxic effect of test compounds (**2–13**) on human neuroblastoma SH-SY5Y cells at concentrations of 10 μM, 50 μM and 100 μM, after 24 h of exposure. Data are presented as mean ± SEM (n = 9). The level of significance is expressed as: * *p* < 0.05, ** *p* < 0.01, *** *p* < 0.001, **** *p* < 0.0001.

**Figure 5 molecules-25-04552-f005:**
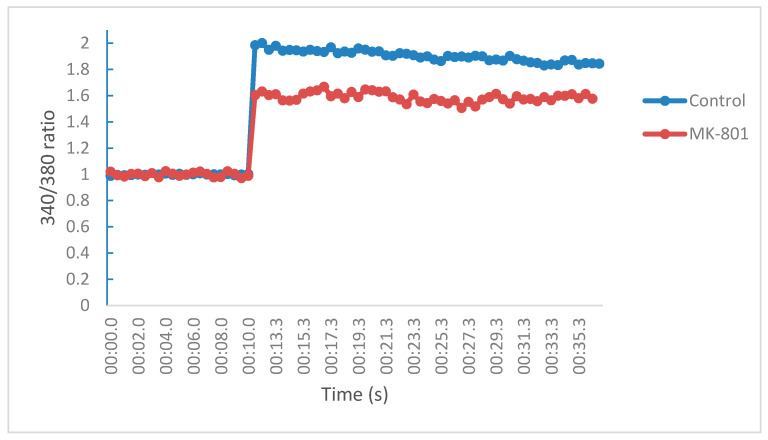
Fluorescence profile of SH-SY5Y neuroblastoma cells in a Fura-2 based assay in the presence or absence of positive control (MK-801) for NMDA receptor mediated Ca^2+^-studies.

**Figure 6 molecules-25-04552-f006:**
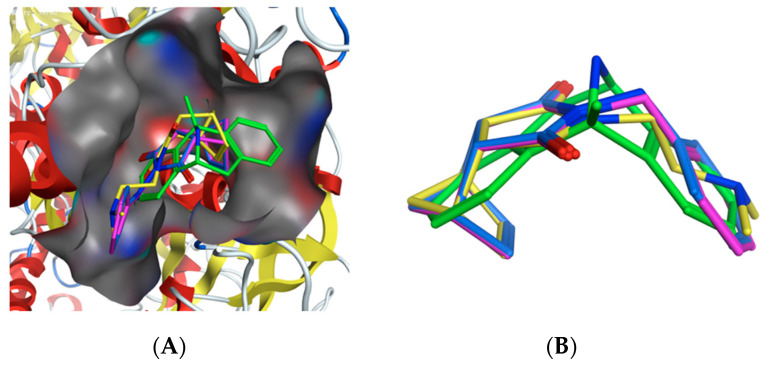
(**A**) Top ranked docking pose of compounds **3** (magenta), **8** (blue) and **13** (yellow). The co-crystallized pose of MK-801 within the NMDA receptor ion channel is shown in green. (**B**) Superposition of MK-801 (green) with compounds **3** (magenta), **8** (blue) and **13** (yellow).

**Figure 7 molecules-25-04552-f007:**
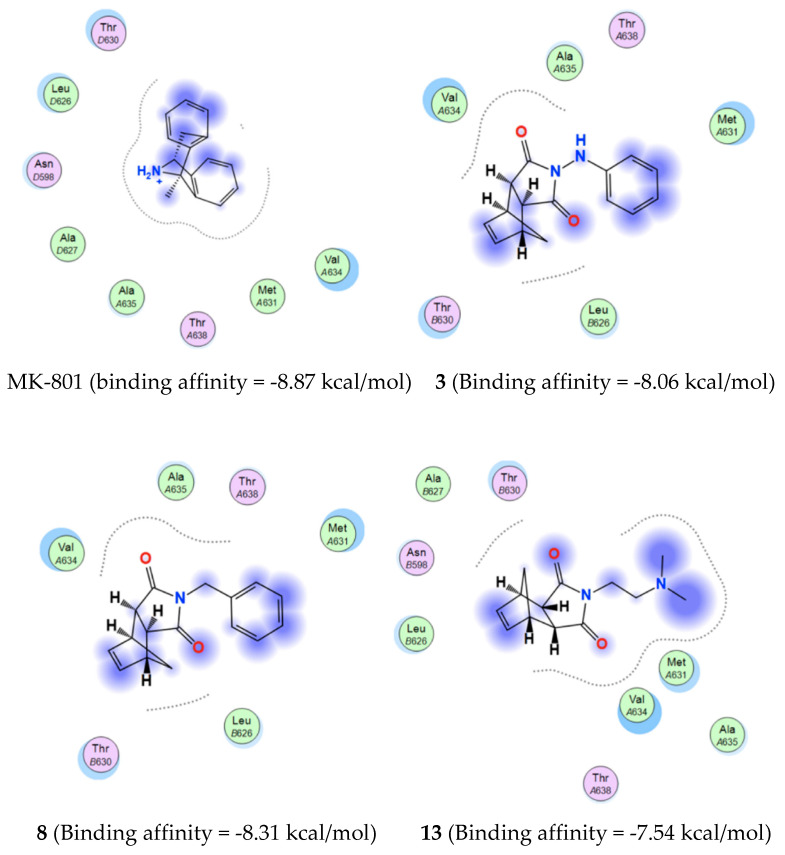
Interaction map of MK-801 and representative compounds **3**, **8** and **13**.

**Figure 8 molecules-25-04552-f008:**
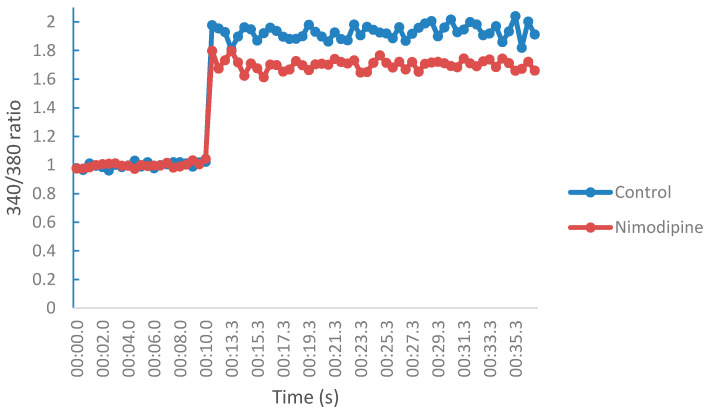
Fluorescence profile of SH-SY5Y neuroblastoma cells in a Fura-2 based assay in the presence or absence of test sample (Nimodipine) for voltage gated Ca^2+^-studies.

**Figure 9 molecules-25-04552-f009:**
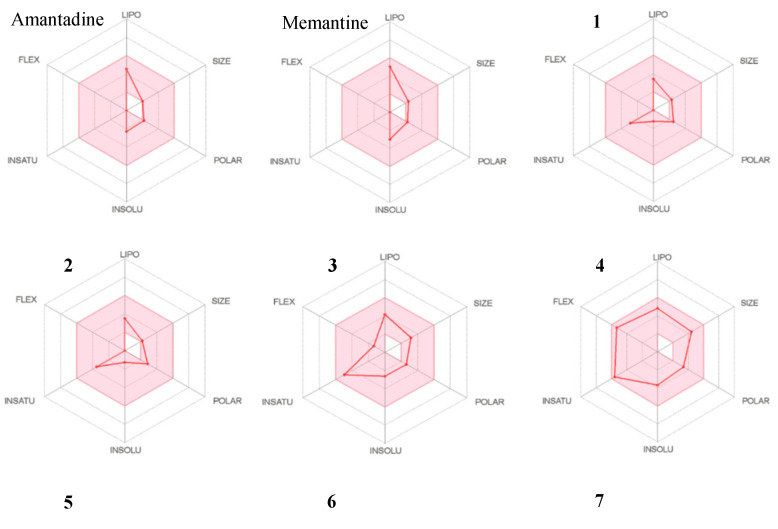
The bioavailability radar chart for amantadine, memantine and compounds **1**–**13**.

**Table 1 molecules-25-04552-t001:** The biological activity profiles of compounds **1**–**13** and related reference compounds.

Compound	Cell Viability %	Neuroprotection % ^a^	VGCC % Inhibition	NMDA % Inhibition
	10 μM	10 μM	10 μM	10 μM
**MK-801**	-	-	Inactive	47.5 ± 2.3 ****
**Nimodipine**	-	-	34.1 ± 4.8 ****	Inactive
**NGP1-01**	86 ± 7.8	65.6 ± 6.9	34.2 ± 4.0 ****	38.1 ± 3.7 ****
**1**	96.6 ± 4.2	40.8 ± 5.6 ****	25.4 ± 2.8 ****	51.0 ± 3.22 ****
**2**	89.0 ± 2.8 *	30.9 ± 3.1 ****	23.1 ± 8.3 ****	23.5 ± 3.5 ****
**3**	84.4 ± 1.8 ***	53.6 ± 9.3 ****	37.2 ± 10.0 ****	43.3 ± 1.7 ****
**4**	108.3 ± 1.4	29.1 ± 3.0 ****	32.2 ± 10.1 ****	42.9 ± 2.0 ****
**5**	115.0 ± 4.2	32.1 ± 8.2 ***	29.9 ± 3.6 ****	43.7 ± 0.5 ****
**6**	103.5 ± 2.1	27.5 ± 6.0 ***	30.2 ± 8.9 ****	43.8 ± 2.2 ****
**7**	114.3 ± 4.9	30.5 ± 0.8 ****	28.0 ± 6.2 ****	53.9 ± 2.1 ****
**8**	95.5 ± 4.4	35.3 ± 5.7 ****	26.8 ± 2.4 ****	45.1 ± 4.3 ****
**9**	95.9 ± 1.4 *	29.9 ± 6.8 ***	22.2 ± 4.9 ****	36.8 ± 3.4 ****
**10**	77.3 ± 3.9 **	27.2 ± 8.6 ***	31.4 ± 5.2 ****	50.4 ± 2.3 ****
**11**	98.1 ± 3.5	35.1 ± 1.1 ****	37.9 ± 9.3 ****	50.4 ± 2.3 ****
**12**	92.8 ± 5.0	28.8 ± 10.4 **	15.4 ± 3.5 ****	50.4 ± 2.3 ****
**13**	99.9 ± 1.8	23.0 ± 3.5 ***	7.5 ± 4.6 **	64.3 ± 1.2 ****

^a^ Percentage neuroprotection values calculated as the difference between the final percentage cell viability of the test compound treated cell line and that of the 2 mM MPP^+^ only treated cell line. Data are presented as mean ± SEM (n = 9). Statistical analysis was performed on raw data, with asterisks indicating significant activity (* *p* < 0.05, ** *p* < 0.01, *** *p* < 0.001, **** *p* < 0.0001).

**Table 2 molecules-25-04552-t002:** Binding affinities and experimental inhibition constants (Ki) values of the known NMDA receptor antagonists.

Compound	Binding Affinity (kca·mol^−1^)	Experimental Ki (nM)
MK-801	−8.87	37.2^34^
Phencyclidine	−8.18	59^35^
Memantine	−6.51	540^36^
R-Ketamine	−6.42	659^35^
Amantadine	−5.25	10,000^37^

**Table 3 molecules-25-04552-t003:** In silico ADME and drug-likeness properties of memantine, amantadine and compounds **1**–**13**.

Compounds	GIA ^a^	BBB ^b^	P-gp ^c^	Drug-likeness					
				Lipinski	Ghose	Veber	Egan	Muegge	Bio. Score ^d^
Amantadine	High	Yes	No	Yes	No ^e^	Yes	Yes	No ^f^	0.55
Memantine	High	Yes	No	Yes	Yes	Yes	Yes	No ^f^	0.55
**1**	High	Yes	No	Yes	Yes	Yes	Yes	No ^e^	0.55
**2**	High	Yes	No	Yes	Yes	Yes	Yes	No ^e^	0.55
**3**	High	Yes	No	Yes	Yes	Yes	Yes	Yes	0.55
**4**	High	Yes	Yes	Yes	Yes	Yes	Yes	Yes	0.55
**5**	High	Yes	No	Yes	Yes	Yes	Yes	Yes	0.55
**6**	High	No	No	Yes	Yes	Yes	Yes	Yes	0.55
**7**	High	Yes	No	Yes	Yes	Yes	Yes	Yes	0.55
**8**	High	Yes	No	Yes	Yes	Yes	Yes	Yes	0.55
**9**	High	No	No	Yes	No ^e^	Yes	Yes	Yes	0.55
**10**	High	No	Yes	Yes	Yes	Yes	Yes	Yes	0.55
**11**	High	Yes	No	Yes	Yes	Yes	Yes	Yes	0.55
**12**	High	Yes	No	Yes	Yes	Yes	Yes	Yes	0.55
**13**	High	Yes	No	Yes	Yes	Yes	Yes	Yes	0.55

^a^ GIA: Gastrointestinal absorption; ^b^ BBB: Blood brain barrier permeant; ^c^ P-gp: P-glycoprotein; ^d^ Bio. Score: Bioavailability score. ^e^ 1 violation; ^f^ 2 violations.

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
