# Peer review of "4-Oxatricyclo[5.2.1.02,6]dec-8-ene-3,5-dione Derivatives as NMDA Receptor- and VGCC Blockers with Neuroprotective Potential"

_molecules, 2020, doi:10.3390/molecules25194552_

Round 1
Reviewer 1 Report
General Comments:
The manuscript by Egunlusi et al, has as objective to do a molecular screening to determine the neuroprotective effect of tricycloundecene derivatives, on the NMDA receptor and VGCC in SHSY5Y neuroblastoma cells. The compounds derivatives demonstrated mono or dual calcium modulating effect on the NMDA receptor and/or VGCC. Molecular docking shown NMDA receptor bind in a comparable manner to the crystallographic pose of MK-801 inside the NMDA ion channel.
Suggestions
1) In line 357 the authors need to verify the final concentration and final volume of MTT in the cytotoxicity assay.
2) Authors should include the description of statistical methods.
3) Authors should show all controls (negative, positive and vehicle): NMDA, 770mM KCl, MK-801 and Nimodipine in the neuroprotection assays.
4) Authors should show the results of fluorometric calcium determinations for each molecule with all controls (negative, positive and vehicle).
5) The figure 6 needs to verify the binding affinity values.
6) The figure 7 needs to adjust the numbers vs each radar figure because appear misaligned.
7) It would be interesting if the authors can show the binding affinities from docking constants of all molecules.
Specific Comment:
In the lines 109-112 the authors tell that the increase cell viability of the molecules 3, 6, and 10 maybe due to these molecules being able to reduce apoptosis. However, in the manuscript there not are any experimental assay to demonstrate this cue. How are you sure of that? Can be possible a proliferative effect?
Author Response
* In line 357 the authors need to verify the final concentration and final volume of MTT in the cytotoxicity assay.
Author’s response:
The final concentration and volume of MTT in the cytotoxicity assay were 0.5 mg/ml and 100 µl, respectively. A statement regarding this was included.
*Authors should include the description of statistical methods.
Author’s response:
The GraphPad Prism software was used. ANOVA and t-test analyses were conducted. A section for statistical analysis was included.
*Authors should show all controls (negative, positive and vehicle): NMDA, 770mM KCl, MK-801 and Nimodipine in the neuroprotection assays
Author’s response:
Data are not available for these compounds in the neuroprotection assays. The study focused on the ability of synthesised compounds to inhibit MPP+-induced toxicity. Moreover, no study has demonstrated neuroprotective effects, through attenuating MPP+-induced toxicity, of these compounds (NMDA, KCl, MK-801 and Nimodipine), thus their activities were not investigated.
*Authors should show the results of fluorometric calcium determinations for each molecule with all controls (negative, positive and vehicle)
Author response:
The values in the table summarised the calcium inhibitory profile of all compounds. It was analysed and determined from numerous data set of 340nm/380nm fluorescence reading. However, we include the fluorescence profile of control (vehicle), Nimodipine and MK-801 as examples.
*The figure 6 needs to verify the binding affinity values.
Author response:
The binding affinity values in figure 7 (initially 6) have been verified.
*The figure 7 needs to adjust the numbers vs each radar figure because appear misaligned.
Author response:
Figure 7 was changed to figure 9 due to additional figures, and the numbers vs radar have been aligned as advised.
*It would be interesting if the authors can show the binding affinities from docking constants of all molecules.
Author response:
The binding affinity values for the compounds of interest were supplied in Figure 7 and Table 2 and the presence of an additional table with all the values will not add much value to the article, in our opinion.
*In the lines 109-112 the authors tell that the increase cell viability of the molecules 3, 6, and 10 maybe due to these molecules being able to reduce apoptosis. However, in the manuscript there not are any experimental assay to demonstrate this cue. How are you sure of that? Can be possible a proliferative effect?
Author response:
The study focused on the cytotoxic profile of these compounds. The claim was a suggestion and will need further experiment to demonstrate. It could possibly be a proliferative effect as suggested and statement regarding this has been included.
Reviewer 2 Report
The authors of the manuscript entitled “4-Oxatricyclo[5.2.1.0 2,6]dec-8-ene-3,5-dione derivatives as NMDA receptor- and VGCC blockers with neuroprotective potential” by Egunlusi et al. demonstrate a dozen compounds that may be potential NMDAR and VGCC inhibitors. Therefore, they could act as neuroprotective agents in neurons, protecting against unfavorable overload of cells with calcium ions.
This is an interesting topic that will be of interest to molecules readers and investigators studying the function of NMDA receptors and VGCCs. In general the experiments have been well carried out and the manuscript is well written. However, there are several issues that need to be resolved before this work can be published:
Some of the major concerns include:
- Table 1: The table presents the research results very nicely. However, charts from these experiences are missing. Please attach Fura340 / Fura 380 flow charts to observe, for example, whether the channels open just as quickly for each compound or whether the channels then close after the same time.
- Table 1 and methods: Did the authors see an increase in the concentration of Ca2+ ions during Ca2+ influx experiments by adding only 100 µM CaCl2 concentration (in the presence of NMDA and glycine)? Usually, when testing the influx of Ca2 + ions, they are added in a concentration of about 2 mM.
- Table 1: What exactly was measured and how the inhibition percentages were calculated? Please state whether these were the maximum peak values or the area under the curve was taken.
- SH-SY5Y cells contain also e.g. AMPA or KA receptors. By activating NMDA receptors, the influx of Ca2+ through AMPA receptors is also possible. Why was no inhibitor of this receptor added in the experiments? Was it checked before? Please explain.
- Data are presented as mean+/- SEM (n=9). What does the “n” mean? Number of cells? If so, there are not enough of them. If not, please provide the number of cells analyzed.
- Does the 10 mM nimodipine concentration is not too high? Typically a concentration of the order of 5-20 µM is used. It is recommended to repeat the experiments. Please explain.
- Figure 7: Numbers for compounds 11-13 are missing on the figure. Moreover, they are in the wrong place. The chart seems to have "gone".
- Please explain why the authors choose SH-SY5Y cells for this study and not neurons, which are the best model for studying, for example, neuroprotection?
- Please summarize whether, on the basis of all the results obtained, one or more compounds that work best can be selected (or are all equally good).
Some of the minor concerns include:
- Line 381: please specify the composition of the Krebs buffer
- There are several typos and a through proof reading is required. Below are few examples:
Line 85: “cyctotoxic” should be “cytotoxic”
Line 106, 147, 276, 280: “Data is” should be “data are” as in line 361
Line 171: „SY-SY5Y” should be “SH-SY5Y”
Author Response
* Table 1: The table presents the research results very nicely. However, charts from these experiences are missing. Please attach Fura 340 / Fura 380 flow charts to observe, for example, whether the channels open just as quickly for each compound or whether the channels then close after the same time.
Author response;
The Fura 340/380 charts for the NMDA receptor and VGCC mediated studies have been included as figure 5 and figure 8, respectively. The charts were described in text.
* Table 1 and methods: Did the authors see an increase in the concentration of Ca2+ ions during Ca2+ influx experiments by adding only 100 μM CaCl2 concentration (in the presence of NMDA and glycine)? Usually, when testing the influx of Ca2+ ions, they are added in a concentration of about 2 mM.
Author response:
In this study, calcium containing buffers (2 mM CaCl2) were added to each well during sample preparation. The addition 100 µM CaCl2 concentration was included in the depolarising buffers and total calcium concentration was enough to alter the fluorescence behaviour of the cells as observed in the Fura flow charts (figure 5 and 8). We have included the concentration of calcium containing buffers for clarity.
* Table 1: What exactly was measured and how the inhibition percentages were calculated? Please state whether these were the maximum peak values or the area under the curve was taken.
Author response:
The fluorescence ratios (340/380) versus time obtained in the presence or absence of test samples were plotted on a graph. From the graph, linear equations were derived and the estimated calcium influx values for control and test samples, at 36.8 seconds, were extrapolated. Relative to the control (DMSO), the inhibition percentages were calculated using the following formula: (control-test compound/control) x 100. An explanation on how the inhibition percentages were calculated has been included.
* SH-SY5Y cells contain also e.g. AMPA or KA receptors. By activating NMDA receptors, the influx of Ca2+ through AMPA receptors is also possible. Why was no inhibitor of this receptor added in the experiments? Was it checked before? Please explain.
Author response:
In the NMDA receptor mediated Ca2+ studies, NMDA (not glutamate) was used as the agonist. NMDA is a widely known agonist specific for NMDA receptors, thus the exclusion of AMPA and KA inhibitors (D’Aniello et al., 2007). . However, it will be interesting to see the impact of AMPA and KA inhibitors on the calcium inhibitions if glutamate is used in future studies.
Reference
D’Aniello, S., Fisher, G., Topo, E., Ferrandino, G., Garcia-Fernandez, J. & D’Aniello, A. (2007) N-Methyl-D-aspartic acid (NMDA) in the nervous system of amphioxus Branchiostoma lanceolatum. BMC neuroscience, 8 (1), 109.
* Data are presented as mean+/- SEM (n=9). What does the “n” mean? Number of cells? If so, there are not enough of them. If not, please provide the number of cells analyzed.
Author response:
The ‘n’ is not the number of cells but the number of experiments. Each experiment contains a series of data points. The values reported in Table 1 were data from graphs plotted on GraphPrism statistical software. The number of cells in each well was in the order of 1.0 x 104
* Does the 10 mM nimodipine concentration is not too high? Typically, a concentration of the order of 5-20 μM is used. It is recommended to repeat the experiments. Please explain.
Author response:
10 µM of Nimodipine was used in the experiment as evident in Table 1. The 10 mM observed in text was a topographical error, which has been corrected accordingly.
* Figure 7: Numbers for compounds 11-13 are missing on the figure. Moreover, they are in the wrong place. The chart seems to have "gone".
Author response:
Figure 7 was renamed due to additional figures, and all numbers have been correctly allocated to radars.
*Please explain why the authors choose SH-SY5Y cells for this study and not neurons, which are the best model for studying, for example, neuroprotection?
Author response:
SH-SY5Y cells are known to exhibit physiological and pathological similarities with primary neurons, thus it usage to screen known and unknown calcium blockers. Additionally, NMDA induced SH-SY5Y cell injury model is believed to mimic glutamate-induced toxicity in human neurons and is generally accepted as activity screening model for the neuroprotective effect of NMDA receptor inhibitors (Warnock et al, 2013; Liu et al., 2019).
These statements with references were included in the article. Moreover, the use of primary neurons is limited due to their ethical concerns.
References
Warnock, A., Tan, L., Li, C., Haack, K., Narayan, S. & Bennett, M. (2013) Amlodipine prevents apoptotic cell death by correction of elevated intracellular calcium in a primary neuronal model of batten disease (CLN3 disease). Biochemical and Biophysical Research Communications, 436, 645-649.
Liu, J., Fan, Y., Kim, D., Zhong, T., Yi, P., Fan, C., Wang, A., Yang, X., Lee, S., Ren, X. & Xu, Y. (2019) Neuroprotective effect of catechins derivatives isolated from Anhua dark tea on NMDA-induced excitotoxicity in SH-SY5Y cells. Fitoterapia, 137, 104240. Available at https://doi.org/10.1016/j.fitote.2019.104240: accessed January 23, 2020.
*Please summarize whether, on the basis of all the results obtained, one or more compounds that work best can be selected (or are all equally good).
Author response:
Compared to the known blockers, all the compounds exhibit good activities. This was already summarised in the conclusion. In general, all the compounds had activity in the same order of magnitude.
*Line 381: please specify the composition of the Krebs buffer
Author response:
The Krebs buffer solution contains 118 mM NaCl, 4.7 mM KCl, 20 mM HEPES, 30.9 mM glucose monohydrate. This was included in the article.
*There are several typos and a through proof reading is required. Below are few examples:
Line 85: “cyctotoxic” should be “cytotoxic”
Line 106, 147, 276, 280: “Data is” should be “data are” as in line 361
Line 171: „SY-SY5Y” should be “SH-SY5Y”
Author response:
The article was proof read and all grammatical errors were corrected as advised
Round 2
Reviewer 1 Report
The authors made the suggested adjustments in the presentation of the results and in the text to improve the manuscript. In my opinion the article can be published as it is presented now.
Reviewer 2 Report
The authors of the manuscript entitled “4-Oxatricyclo[5.2.1.0 2,6]dec-8-ene-3,5-dione derivatives as NMDA receptor- and VGCC blockers with neuroprotective potential” by Egunlusi et al. demonstrate a dozen compounds that may be potential NMDAR and VGCC inhibitors. Therefore, they could act as neuroprotective agents in neurons, protecting against unfavorable overload of cells with calcium ions.
This is an interesting topic that will be of interest to molecules readers and investigators studying the function of NMDA receptors and VGCCs.
The authors have satisfied most of my concerns. The MS was improved and in my opinion this revised manuscript can be now published.